# The risk of SARS-CoV-2 outbreaks in low prevalence settings following the removal of travel restrictions

Rahil Sachak-Patwa[1], Helen M. Byrne [1], Louise Dyson [2,3] & Robin N. Thompson [2,3✉]

## Abstract

**Background** Countries around the world have introduced travel restrictions to reduce SARS-CoV-2 transmission. As vaccines are gradually rolled out, attention has turned to when travel restrictions and other non-pharmaceutical interventions (NPIs) can be relaxed.

**Methods** Using SARS-CoV-2 as a case study, we develop a mathematical branching process model to assess the risk that, following the removal of NPIs, cases arriving in low prevalence settings initiate a local outbreak. Our model accounts for changes in background population immunity due to vaccination. We consider two locations with low prevalence in which the vaccine rollout has progressed quickly – specifically, the Isle of Man (a British crown dependency in the Irish Sea) and the country of Israel.

**Results** We show that the outbreak risk is unlikely to be eliminated completely when travel restrictions and other NPIs are removed. This general result is the most important finding of this study, rather than exact quantitative outbreak risk estimates in different locations. It holds even once vaccine programmes are completed. Key factors underlying this result are the potential for transmission even following vaccination, incomplete vaccine uptake, and the recent emergence of SARS-CoV-2 variants with increased transmissibility.

**Conclusions** Combined, the factors described above suggest that, when travel restrictions are relaxed, it may still be necessary to implement surveillance of incoming passengers to identify infected individuals quickly. This measure, as well as tracing and testing (and/or isolating) contacts of detected infected passengers, remains useful to suppress potential outbreaks while global case numbers are high.

### Plain language summary

The effectiveness of public health measures against COVID-19 has varied between countries, with some experiencing many infections and others containing transmission successfully. As vaccines are deployed, an important challenge is deciding when to relax measures. Here, we consider locations with few cases, and investigate whether vaccination can ever eliminate the risk of COVID-19 outbreaks completely, allowing measures to be removed risk-free. Using a mathematical model, we demonstrate that there is still a risk that imported cases initiate outbreaks when measures are removed, even if most of the population is fully vaccinated. This highlights the need for continued vigilance in low prevalence settings to prevent imported cases leading to local transmission. Until case numbers are reduced globally, so that SARS-CoV-2 spread between countries is less likely, the risk of outbreaks in low prevalence settings will remain.

[1] Mathematical Institute, University of Oxford, Oxford, UK. [2] Mathematics Institute, University of Warwick, Coventry, UK. [3] Zeeman Institute for Systems Biology and Infectious Disease Epidemiology Research, University of Warwick, Coventry, UK. ✉email: robin.n.thompson@warwick.ac.uk

Combinations of non-pharmaceutical interventions (NPIs) have been introduced worldwide to counter the COVID-19 pandemic[1–3]. These measures include travel restrictions[4,5] and a range of other NPIs intended to reduce the number of contacts between individuals[6]. The development and deployment of vaccines have also lowered transmission[7–10] and reduced the number of individuals experiencing clinical symptoms or severe disease once infected[11–13].

Effective NPIs have led to low levels of transmission in some locations. Australia and New Zealand have often been cited as examples of countries that have implemented NPIs effectively, with travel restrictions and quarantine of inbound travellers combined with short-term lockdowns and contact tracing to identify infected contacts whenever cases have been discovered[14–17]. Rigorous interventions targeted at inbound travellers in low prevalence settings reflect the fact that imported cases can contribute substantially to local incidence[18], and potentially initiate outbreaks with substantial local transmission[19,20]. At the time of writing (1st May 2021), Israel is the country that has vaccinated the largest proportion of its citizens, and the vaccination campaign there has been credited with reducing transmission[8], prompting some NPIs to be removed.

Despite the success of both NPIs and vaccines, the current overall picture is complicated. Vaccines do not prevent transmission entirely[7,21–24], and vaccine uptake is incomplete, particularly in some ethnic groups and in underserved communities[25]. Current tentative estimates suggest that first doses of the Pfizer and AstraZeneca SARS-CoV-2 vaccines are around 60% effective at preventing infection, with second doses leading to 65-85% effectiveness against infection[21]. Furthermore, the appearance of novel SARS-CoV-2 variants makes eliminating transmission more challenging. For example, the Alpha variant (B.1.1.7, or VOC 202012/01) that first appeared in the United Kingdom in late 2020 has been found to be more transmissible than the SARS-CoV-2 virus that originally emerged in China[26]. Although public health measures have had some successes, these concerns raise a key question: Can NPIs such as travel restrictions be removed without any risk in low prevalence settings where vaccines have been distributed widely, or might outbreaks occur initiated by SARS-CoV-2 reimportations from elsewhere?

Epidemiological models are often used to assess the risk of outbreaks in scenarios in which the potential for pathogen transmission is not changing. According to the mathematical theory of branching processes, the probability that cases introduced into a new host population generate an outbreak driven by sustained local transmission is then greater than zero whenever $R > 1$, where $R$ is the reproduction number of the pathogen. For the most common branching process model, used to represent the spread of directly transmitted pathogens, the probability that $I_0$ introduced infectious cases initiate an outbreak is given by

$$\text{Prob(outbreak)} = 1 - \left(\tfrac{1}{R}\right)^{I_0}. \quad (1)$$

This expression had been used to assess the risks of outbreaks of pathogens including the Ebola virus[27–29], before being applied early in the COVID-19 pandemic to assess outbreak risks outside China[20]. Eq. (1) is based on a simple transmission model in which individuals are assumed to mix homogeneously and infected individuals are infectious for exponentially distributed time periods. Equation (1) also involves an assumption that pathogen transmissibility is fixed at its current level. In other words, the value of $R$ is implicitly assumed to remain constant over the initial phase of the potential outbreak. With a background of rapidly changing population immunity due to vaccination acting to reduce transmission, this assumption may not be accurate. To assess the risk of outbreaks if NPIs are removed during an ongoing vaccination campaign using branching process models, standard epidemiological modelling theory must be extended to account for temporally changing population immunity.

Here, we use branching processes to investigate whether an introduced case is likely to initiate an outbreak, accounting fully for temporal changes in population immunity due to an ongoing vaccination campaign. We use four metrics to assess the risk of an outbreak and consider two examples of vaccination campaigns from around the world, from the Isle of Man and the country of Israel. In both locations, vaccination is progressing quickly and prevalence is currently low. Given the relatively low numbers of cases in these locations during the pandemic, there is also likely to be a background of limited immunity from previous infections. We assess the risk of outbreaks in these places when travel restrictions and other NPIs are removed, considering scenarios in which NPIs are removed at different stages of the vaccination campaigns. Our goal is not to develop a detailed epidemiological model for estimating the outbreak risk precisely. Instead, we use a simple model to show that, even when all vaccines have been deployed in low prevalence settings (i.e. a maximum vaccination proportion $\nu$ of individuals have been vaccinated, where the precise value of $\nu$ varies between locations), the combination of incomplete vaccine uptake, imperfect vaccination, and variants of concern means that the risk of outbreaks due to imported cases will not be eliminated completely when NPIs are removed. This highlights the need for careful monitoring of imported cases until global prevalence is reduced to low levels. Until vaccines are rolled out worldwide, there is still a risk of local transmission arising in low prevalence settings initiated by importations from elsewhere.

## Methods

**Epidemiological model.** We performed our main analyses using a stochastic branching process model that describes virus transmission in the initial stages of a potential outbreak. In this model, following the arrival of a case in the local population, each infected individual generates new infections at rate $\beta(1 - \Lambda(t))$ and infected individuals have a mean infectious period of $1/\mu$ days (Fig. 1a). The times between successive events in the model are assumed to follow exponential distributions. The function $\Lambda(t)$ reflects the extent to which transmission has been reduced by vaccination, where a value of $\Lambda(t) = 0$ corresponds to an entirely unvaccinated population. We consider scenarios in which travel restrictions and other NPIs are removed, and transmission is only limited by vaccine-acquired immunity.

To model an ongoing vaccination campaign, we set $\Lambda(t) = \eta_1 V_1(t - \alpha) + \eta_2 V_2(t - \alpha)$, where $V_1(t)$ is the proportion of individuals in the population who have received a single vaccine dose at time $t$ and $V_2(t)$ is the proportion of individuals in the population who have received two vaccine doses at a time $t$ (Fig. 1b). The parameters $\eta_1$ and $\eta_2$ reflect the effectiveness of the vaccine at preventing infection after one and two effective doses, respectively (specifically, $\eta_1$ and $\eta_2$ represent the multiplicative reduction in susceptibility following one or two effective doses, compared to an unvaccinated individual, so that $0 \le \eta_1 \le \eta_2 \le 1$), and the parameter $\alpha$ represents the delay between a vaccine dose being administered and being effective in the recipient. In our main analyses, since we are modelling relatively low prevalence settings, we do not consider immunity due to prior infections, although we present a supplementary analysis in which we demonstrate the robustness of our results to this assumption.

Under this model, the time-dependent reproduction number, accounting for any vaccines that have been administered and are

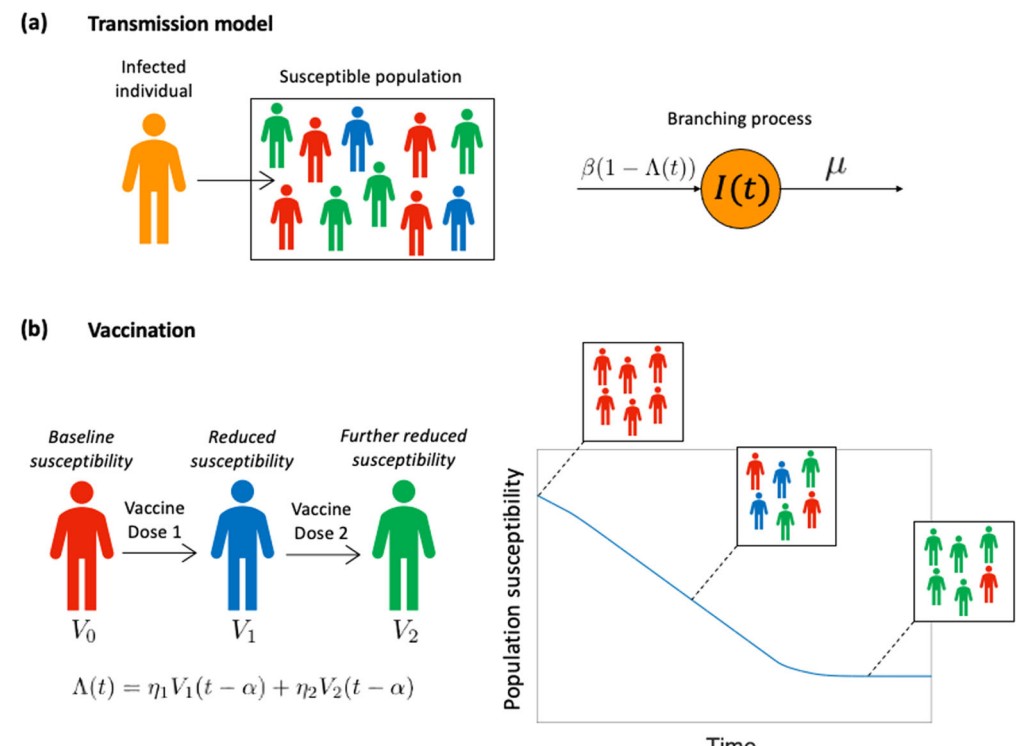

**Fig. 1 The epidemiological model used in our analyses. a** Following the introduction of an infected individual into the host population, local transmission may happen with each infected individual generating new infections at rate $\beta(1 - \Lambda(t))$, where $\Lambda(t)$ reflects the vaccination coverage in the local population. The mean infectious period of infected individuals is $1/\mu$ days, and the rates shown are per infected individual. **b** The vaccination process is modelled by setting $\Lambda(t)$ according to the proportion of individuals in the population who have been vaccinated with one ($V_1$) or two ($V_2$) doses. The first vaccine dose is assumed to have effectiveness $\eta_1$ and the second vaccine dose has effectiveness $\eta_2$, where the values of $\eta_1$ and $\eta_2$ are the multiplicative reductions in the susceptibility of individuals who have been vaccinated with one or two vaccine doses, respectively (compared to an unvaccinated host). Vaccine doses are effective $\alpha$ days after they are administered. This leads to declining population susceptibility as a vaccine is rolled out across the population.

effective at time $t$, is given by

$$R_V(t) = \frac{\beta(1 - \Lambda(t))}{\mu}. \qquad (2)$$

The value of $R_V(t)$ represents the expected number of secondary infections generated by a single infected individual in the population at time $t$, under the assumption that no further vaccinations take place in the future. It is therefore representative of the "instantaneous" transmissibility at time $t$. Indeed, $R_V(t)$ is sometimes referred to as the instantaneous reproduction number[30–33]. In the absence of vaccination, so that $\Lambda(t) = 0$, then $R_V(t)$ is equal to the basic reproduction number, $R_0$.

**Vaccination data**. In the model, $V_1(t)$ and $V_2(t)$ were set based on vaccination data from the location under consideration (either the Isle of Man or Israel). Data describing the proportion of the total population who had received one or two vaccine doses were available for the periods up until 11th April 2021 (for the Isle of Man[34]) and 21st April 2021 (for Israel[35]).

To explore how the risk of outbreaks is likely to change in the future, we projected the vaccine rollout forwards beyond these dates in the following way. We considered the total population size of the location under consideration (denoted $N$), as well as the numbers of individuals ($N_1(t)$ and $N_2(t)$) vaccinated with one or two doses, so that $V_1(t) = N_1(t)/N$ and $V_2(t) = N_2(t)/N$. We assumed that a constant number of vaccine doses are available each day in the future (denoted $D$), and that there is a target period of $\tau$ days between each vaccine dose. On any day in the future, each available dose is assigned to an individual who has been vaccinated with their first dose at least $\tau$ days ago, with

remaining doses then assigned to unvaccinated individuals. Resulting values of $N_1(t)$ and $N_2(t)$ were then converted to corresponding values of $V_1(t)$ and $V_2(t)$. To reflect the fact that vaccine uptake is imperfect, and some population groups are not vaccinated, we assumed that a maximum proportion $\nu$ of the population can ever be vaccinated. Consequently, once the vaccination programme is completed, then $\Lambda(t) = \eta_2\nu$. The value of $\nu$ varies between locations, and depends on factors such as the age distribution of the host population. Since there is uncertainty in the vaccine uptake going forwards, we conducted supplementary analyses in which we considered a range of different values of $\nu$. Values of the model parameters used in the analyses in the main text for the Isle of Man and Israel are shown in Table 1.

**Outbreak risk metrics**. We used four metrics to assess the risk that an infected individual, introduced into the population at time $t$, initiates an outbreak driven by sustained local transmission. The values of each of these metrics vary temporally. Each metric is used to estimate the risk that an imported case initiates an outbreak under the assumption that all travel restrictions and other NPIs are removed (in other words, pathogen transmissibility at time $t$ is determined by $R_V(t)$, which represents an adjustment to the basic reproduction number $R_0$ due to vaccine-acquired immunity, as described above). An overview of the four metrics explored is provided here; additional details are available in the Supplementary Information (Supplementary Methods).

The first metric we considered is the *Instantaneous Outbreak Risk (IOR)*. This quantity represents the expression shown in Eq. (1), with $R = R_V(t)$ and $I_0 = 1$. The IOR reflects the risk of an

**Table 1 Default parameter values used in our analyses.**

| Parameter | Description | Value | Source |
|---|---|---|---|
| $\beta$ | Transmission rate | Set so that $R_0 = \frac{\beta}{\mu}$ takes a prescribed value | Two scenarios were considered, with mean $R_0$ values of 3 (similar to the original SARS-CoV-2 variant[65]) and 5 (similar to the Alpha variant (B.1.1.7)[26]) |
| $1/\mu$ | Duration of infectiousness | 5 days | [66] |
| $\alpha$ | Delay between a vaccine dose being administered and being effective | 14 days | [67] |
| $\eta_1$ | Multiplicative reduction in susceptibility of an individual vaccinated with a single effective dose (compared to an unvaccinated host) | 0.6 | [21] |
| $\eta_2$ | Multiplicative reduction in susceptibility of an individual vaccinated with two effective doses (compared to an unvaccinated host) | 0.85 | [21] |
| $\nu$ | Vaccine uptake | 0.8 (Isle of Man) and 0.7 (Israel). Other values are considered in the Supplementary Information | [68] |
| $N$ | Total population size | 84,500 (Isle of Man) and 8,772,800 (Israel) | [69] |
| $D$ | Number of vaccine doses available each day (in future projections) | 1000 (Isle of Man) and 56,000 (Israel) | Average number of doses administered each day in the previous 30 days (up to 11th April 2021 for Isle of Man[34] and 21st April 2021 for Israel[35]) |
| $\tau$ | Target period between vaccine doses (in future projections) | 84 days (Isle of Man) and 21 days (Israel) | [70,71] |

outbreak occurring starting from a single infected individual at time $t$, but under the assumption that the vaccine rollout does not continue after time $t$ so that pathogen transmissibility is unchanged. While the IOR straightforward to calculate, it does not reflect changing population immunity due to vaccination over the initial phase of the potential outbreak. This standard metric is often used to assess the risk of outbreaks in scenarios where pathogen transmissibility does not vary temporally [20,27–29,36,37].

The second metric is the *Case Outbreak Risk (COR)*. The COR is an extension of the IOR, accounting for changes in population susceptibility due to vaccination over the initial phase of the potential outbreak. The COR has previously been used to assess outbreak risks using branching processes for models in which pathogen transmission varies periodically[38–40]. Its calculation involves solving the differential equation

$$\frac{dq(t)}{dt} = \beta(1 - \Lambda(t))q(t)(1 - q(t)) + \mu(q(t) - 1). \quad (3)$$

The COR at time $t$ is then given by $1 - q(t)$, where $q(t)$ represents the probability of an outbreak failing to occur (i.e. ultimate extinction of the virus in the branching process model[41]) starting from a single infected individual introduced into the population at time $t$. Further details, including the derivation of Eq. (3), are provided in the Supplementary Information (Supplementary Methods).

The third metric we considered is the *Simulated Outbreak Risk (SOR)*. The SOR involves repeated simulation of the branching process model, using the direct version of the Gillespie stochastic simulation algorithm[42] adapted to account for temporally varying pathogen transmissibility (due to changing population immunity as vaccines are deployed). Simulations were run starting with a single infected individual introduced into the population at time $t$. The SOR then corresponds to the proportion of simulations in which a local outbreak occurs; an outbreak is said to occur if the total number of individuals infected simultaneously exceeds a pre-defined threshold value, $M$. In our analyses, we set $M = 100$.

Finally, we considered the *Numerical Outbreak Risk (NOR)*. The NOR is analogous to the SOR, but with the advantage that it does not require large numbers of model simulations to be run. The NOR, therefore, also represents the risk that a single infected individual introduced into the population at time $t$ initiates an outbreak in which at least $M = 100$ individuals are ever infected simultaneously.

## Results

As described in the Methods, we first generated projections of the number of vaccinated individuals in the future for the Isle of Man (Fig. 2a) and Israel (Fig. 2d), based on past vaccination data in those locations. To explore the impact of vaccination on virus transmission, we calculated the time-dependent reproduction number ($R_V(t)$; Eq. (2)) throughout the vaccination campaigns. We considered two different scenarios. In the first, we set the mean value of $R_0$ (i.e. the reproduction number in the absence of vaccination) equal to 3, as was the situation early in the COVID-19 pandemic (Fig. 2b,e). In the second scenario, we set the mean value of $R_0$ equal to 5 (Fig. 2c,f) to reflect the fact that currently circulating SARS-CoV-2 variants are more transmissible than the original virus[26]. This second scenario is therefore likely to be a more realistic reflection of the current and future risk.

We then calculated the values of the four different outbreak risk metrics throughout the period considered (18th December 2020 to 20th August 2021) based on these vaccination projections (Fig. 3). This involves a scenario in which NPIs are removed entirely, so that $R_V(t)$ is not reduced by interventions other than vaccination. These metrics then reflect the risk that a single case first entering the population at the date of introduction shown initiates an outbreak driven by sustained local transmission, given that no NPIs are in place. For each metric, and each time during the vaccination programme, we calculated the outbreak risk by integrating over the full distribution for $R_V(t)$ shown in Fig. 2. In practice, the precise value of $R_V(t)$ is not known exactly at any time: the purpose of this integration (which is over values of $R_V(t)$, rather than values of $t$) is to account for this uncertainty in outbreak risk estimates. The resulting outbreak risk therefore

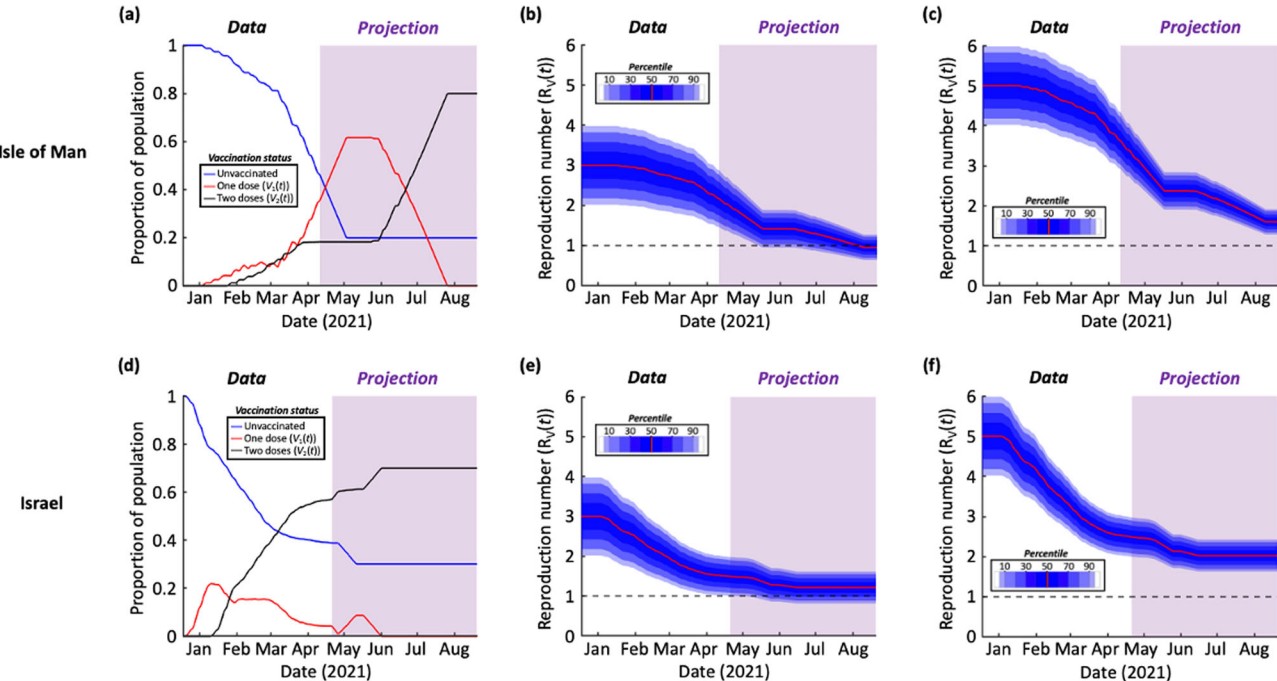

**Fig. 2 Effect of vaccination on population susceptibility. a** The proportion of the population of the Isle of Man that is unvaccinated ($1 - V_1(t) - V_2(t)$), vaccinated with a single dose ($V_1(t)$) and vaccinated with two doses ($V_2(t)$). The period in which vaccination data were available is shown in white, and the period in which vaccination data were projected is shown in purple. **b** The time-dependent reproduction number ($R_V(t)$) corresponding to the vaccination data in panel a with all travel restrictions and non-pharmaceutical interventions removed, starting from a mean initial value of $R_V(0) = R_0 = 3$. To account for uncertainty in the value of $R_0$, a normal distribution was assumed about the mean value of $R_0$ with variance $\sigma^2 = 0.25$, which is reflective of the range of $R_0$ values estimated early in the COVID-19 pandemic[65]. **c** Analogous to panel b but starting from a mean initial value of $R_V(0) = R_0 = 5$. **d-f** Analogous to panels a-c but using vaccination data for Israel. In all panels, $t = 0$ days corresponds to 18th December 2020. Ticks on the x-axes refer to the starts of the months labelled. Parameter values are shown in Table 1.

represents a point estimate of the risk accounting for uncertainty in pathogen transmissibility.

The IOR represents the probability of sustained local transmission based only on the value of $R_V(t)$ at the precise instant that the virus is introduced into the population. When an introduction occurs while vaccines are being deployed, there is a background of decreasing population susceptibility, and so $R_V(t)$ may in fact decrease over the initial stages of a potential outbreak. For that reason, it is unsurprising that the IOR sometimes overestimates the outbreak risk compared to the other risk metrics (see e.g. black lines in Fig. 3a, c – and see Discussion).

In contrast, we found close agreement between the COR, SOR, and NOR. Due to the high assumed vaccine uptake in the Isle of Man, the outbreak risk at the end of the vaccination programme there was calculated to be lower than when the vaccine rollout was completed in Israel (although we also considered supplementary analyses with different assumed vaccine uptake values – Fig. S1). In the first scenario that we considered (mean $R_0 = 3$), which is representative of the transmissibility of the original SARS-CoV-2 virus, the outbreak risk was projected to be low following the vaccination programme in the Isle of Man.

However, when the virus was assumed to be more transmissible (mean $R_0 = 5$), as is the case for newly emerged variants of concern such as the Alpha variant (B.1.1.7), the outbreak risk was found to be substantial even following the projected end of the vaccination campaigns in both the Isle of Man and Israel. For example, when the assumed vaccine uptake values of $\nu = 0.8$ and $\nu = 0.7$ had been achieved in the Isle of Man and Israel, respectively, the NOR took values of 0.37 (95% Equal-Tailed Credible Interval (CrI): [0.22,0.48], calculating using the 95% Equal-Tailed CrI for $R_V(t)$ at the end of the vaccination rollout) and 0.51 (95% Equal-Tailed CrI: [0.39,0.59]) in those locations.

Given the uncertainty in epidemiological modelling projections, the key point to note is that these values are greater than zero (rather than the exact outbreak risk estimates themselves), suggesting that outbreaks can still occur when NPIs are removed following the completion of vaccination programmes. This result is robust to a range of assumptions surrounding virus transmission and the effectiveness of vaccines (Figs S1-S4).

## Discussion

As vaccines are administered in countries around the world, attention has turned to the possibility that transmission will soon be reduced to the extent that travel restrictions and other NPIs can be relaxed. Here, we have investigated the impact of removing NPIs on the risk of outbreaks occurring in locations with low prevalence and a substantial proportion of the population vaccinated. We used four metrics to estimate the risk that a case introduced at any stage in the vaccination rollout leads to an outbreak driven by local transmission, in a scenario in which NPIs are removed. We calculated temporal changes in the values of these metrics in the context of vaccination in the Isle of Man and in Israel, two locations with low prevalence and vaccination campaigns that have progressed quickly.

We found that vaccination is leading to a substantial drop in the potential for virus transmission in both the Isle of Man and Israel, as indicated by a decreasing value of the time-dependent reproduction number, $R_V(t)$ (Fig. 2). However, even when the vaccine rollout is completed (with 80% of individuals fully vaccinated in the Isle of Man, and 70% in Israel; other values are considered in Fig. S1), the combination of vaccines not preventing transmission entirely, incomplete vaccine uptake and the emergence of novel SARS-CoV-2 variants suggests that the risk of

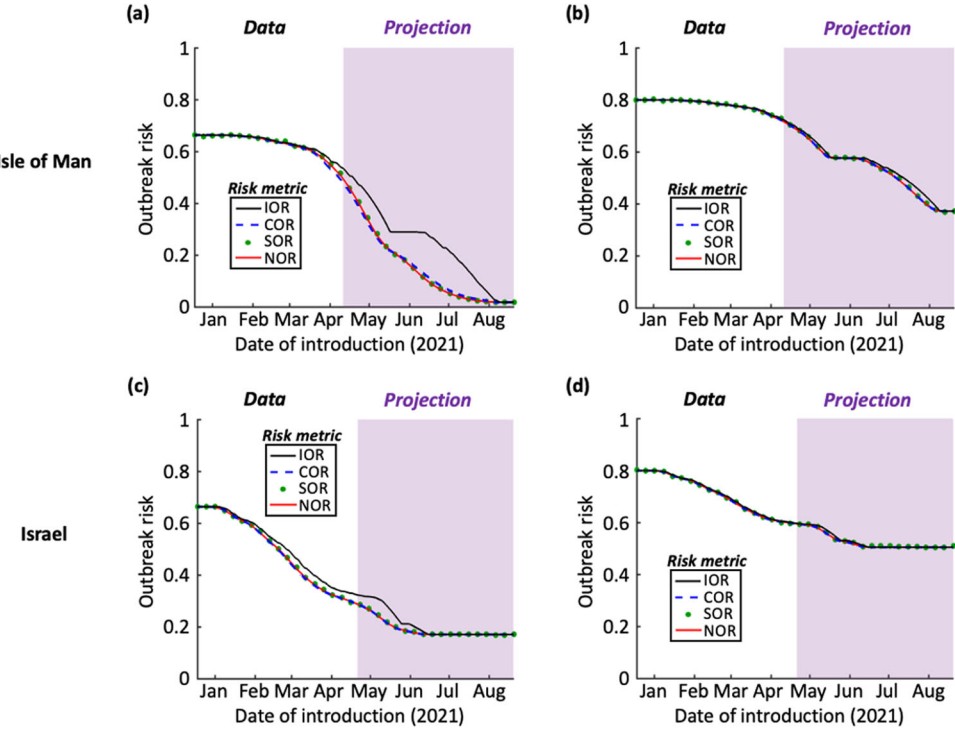

**Fig. 3 The risk that an infectious case introduced at each stage of the vaccination campaign initiates an outbreak, if travel restrictions and other non-pharmaceutical interventions are removed. a** The outbreak risk was assessed using the four metrics (Instantaneous Outbreak Risk – black; Case Outbreak Risk – blue dashed; Simulated Outbreak Risk – green dots; Numerical Outbreak Risk – red), based on vaccination data from the Isle of Man and assuming mean $R_0 = 3$, as in Fig. 2b. The period in which vaccination data were available is shown in white, and the period in which vaccination data were projected is shown in purple. **b** Analogous to panel a but with a mean $R_0 = 5$ (see Fig. 2c). **c** Analogous to panel a but using vaccination data for Israel (see Fig. 2e). **d** Analogous to panel a but using vaccination data for Israel with a mean $R_0 = 5$ (see Fig. 2f). Ticks on the x-axes refer to the starts of the months labelled. Parameter values are shown in Table 1.

outbreaks initiated by infected individuals arriving from elsewhere will not be eliminated when NPIs are removed (Fig. 3). This conclusion holds unless the vaccine uptake is very high (Fig. S1). This suggests that, when NPIs such as travel restrictions are relaxed, it will still be advisable to be aware of the potential for local transmission. Ensuring that case numbers are reduced elsewhere (i.e. in locations that imported cases might travel from) will reduce the risk of importations, and strategies should be considered to suppress outbreaks quickly if importations occur.

One potential use of our modelling framework by policy-makers is to identify the dates on which travel restrictions can be relaxed, based on a maximum acceptable outbreak risk. As an example, if the maximum acceptable value of the NOR is 0.4, then our analysis with $R_0 = 5$ suggests that travel restrictions can be lifted on the Isle of Man at the end of July 2021 (Fig. 3b; the first date on which the NOR is projected to fall below 0.4 in our illustrative analysis is 29th July 2021). However, as described above, since the risk of local transmission following introductions remains, it will be necessary to continue to monitor inbound passengers in low prevalence settings to identify infected individuals, even once vaccination programmes are completed.

Our modelling approach for assessing outbreak risks was motivated by studies in which the potential for pathogen transmission varies periodically[38–40,43–46], due, for example, to seasonal changes in weather conditions that affect transmission. In that context, Carmona and Gandon[38] describe a "winter is coming effect", in which the risk of an outbreak is lower than current environmental conditions suggest if conditions become less favourable for transmission in the near future. In the terminology used in our manuscript, if current environmental conditions promote pathogen transmission, then the IOR is

expected to be high. This is because the IOR reflects the outbreak risk based on the conditions at the precise instance when the virus is introduced into the population. However, if environmental conditions are expected to become less favourable for transmission in the near future, then the values of the other risk metrics are lower than the IOR, since those metrics account for changes in transmissibility over the initial phase of the potential outbreak. Although in general we found a close agreement between the four metrics that we considered, a background of decreasing population susceptibility can lead to a similar effect in which the IOR is larger than the COR, SOR, or NOR (e.g. Figure 3a, c).

In this study, we used a simple branching process model to investigate the risk of outbreaks when NPIs are removed during a vaccination programme. This involved considering whether introduced cases are likely to lead to sustained local transmission or instead fade out without causing an outbreak. We made the standard branching process modelling assumption that population immunity is unaffected by infections in the earliest stages of potential outbreaks[19,20,27–29,47]. In other words, infection-acquired immunity following the arrival of the pathogen in the host population is not considered. While this is reasonable when case numbers are low in the initial stages of potential outbreaks, a more detailed model is needed to explore other quantities, such as the eventual size of outbreaks. Following a vaccination programme, outbreaks are likely to be smaller than those that occur before vaccines are widely administered.

Another simplification of our model is that we only accounted for changes in population susceptibility due to the vaccine rollout. We did not account for prior immunity of some members of the population due to previous exposures to the virus[48]. At the time of writing (1st May 2021), there have been 1,154 confirmed cases

in the Isle of Man and 838,000 confirmed cases in Israel. Since these case numbers correspond to a relatively small proportion of the host population (representing 1.4% and 9.6% of the population in the Isle of Man and Israel, respectively), we do not expect this assumption to affect our key findings. Furthermore, immunity is likely to wane over time[49–51], reducing the effect of previous exposures on the outbreak risk. To test the potential impact of infection-acquired immunity arising from cases occurring before May 2021, we also conducted a supplementary analysis in which the value of $R_0$ is reduced by 1.4% in the Isle of Man and 9.6% in Israel, and we found qualitatively similar results (Fig. S2): even in this scenario, there is still a risk of outbreaks due to imported cases once the vaccination programmes are completed. Importantly, in other countries in which higher numbers of cases have occurred, prior immunity may play a larger role in reducing the risk of outbreaks compared to the low prevalence settings considered here. Understanding the extent of this effect, based on the rate at which immunity wanes, is an important target for further study.

In this research, we assumed that vaccines reduce transmission by lowering the susceptibility of vaccinated hosts compared to unvaccinated hosts. Two effective vaccine doses were assumed to reduce the probability of infection per exposure substantially (by $\eta_2 = 0.85$), which is in line with estimates for the Pfizer vaccine used in the UK Government's COVID-19 roadmap modelling[21]. However, there is uncertainty about the precise level of protection offered by different COVID-19 vaccines. For that reason, we also conducted a supplementary analysis (Fig. S3) in which we considered a range of values of $\eta_2$. In every scenario that we explored, the risk of outbreaks following the completion of vaccination programmes in low prevalence settings was greater than zero, with larger outbreak risks for lower values of $\eta_2$.

In addition to reducing the risk that a host becomes infected as assumed here[52], vaccination may also reduce the risk of onwards transmission following infection[53], reduce the risk of severe disease developing, or a combination of these effects[54]. In principle, it would be possible to develop a more complex model that accounts for all of these effects separately, at the cost of having to estimate (or assume) values of the associated parameter values. Vaccines can be assumed to reduce the risk that vaccinated hosts become infected by lowering their susceptibility to infection compared to unvaccinated hosts (leaky vaccines), or by protecting a subset of the vaccinated hosts completely from infection (all-or-nothing vaccines)[24], and the implications of these different mechanisms on outbreaks while a vaccination campaign is ongoing could be considered. It would also be possible to develop a transmission model in which either vaccine- and/or infection-acquired immunity wanes. There is currently substantial uncertainty about the extent and speed of waning immunity, with the hope that antibody responses and/or cellular immune responses may provide long-lasting immune memory, at least in some individuals[55,56]. Booster vaccination for vulnerable individuals is being considered in high-income countries, given the possibility that vaccine-acquired immunity wanes. We note that any waning of immunity will act to enhance our main qualitative conclusion that the risk of outbreaks is unlikely to be eliminated completely in low prevalence settings by vaccination programmes.

Other factors that may merit consideration in further analyses of future outbreak risks include the impact of population structure on both vaccine effectiveness[57] and transmission[58,59], and the possibility that individuals' behaviours may be different following the pandemic compared to beforehand, even when NPIs and travel restrictions are removed. For precise quantitative outbreak risk predictions to be made, it may be necessary to estimate $R_0$ in different locations. It would also be necessary to incorporate changes in $R_0$ due to further new variants in our analyses, although we note that any increase in transmissibility will act to strengthen our main conclusion that vaccine-acquired immunity alone is unlikely to be sufficient to prevent outbreaks. Another important consideration is the degree of heterogeneity in transmission between different infected individuals. For a range of pathogens, including SARS-CoV-2, the majority of transmissions are generated by a relatively small proportion of infectious hosts[60–63]. We therefore also conducted a supplementary analysis (Supplementary Notes) in which we considered the dependence of outbreak risk estimates following the completion of vaccination programmes in the Isle of Man and Israel on the assumed individual-level variation in SARS-CoV-2 transmission. While a higher degree of heterogeneity (i.e. a higher probability of "superspreading" events) acts to lower the outbreak risk, in each scenario explored outbreaks were still possible in low prevalence settings following vaccination programmes when NPIs are removed (Fig. S4).

To summarise, our analyses have demonstrated the general principle that, even following vaccination programmes in low prevalence settings, the risk of outbreaks remains when NPIs are removed. Our intention is not to argue that travel restrictions and other NPIs should not be relaxed once vaccination programmes are sufficiently advanced, but rather that measures should be taken to ensure that any clusters of cases are suppressed quickly if they arise. A local outbreak requires two steps: first, the virus must be imported from elsewhere; second, local transmission must occur. The first step emphasises the need for a global approach to minimising transmission, since higher case numbers at a potential source location translate into a higher importation risk. The second step emphasises the need to ensure that, while vaccination acts to reduce transmission substantially, continued surveillance of inbound passengers for infection, combined with isolation and/or testing of contacts of detected infected individuals, is important when travel restrictions and other NPIs are relaxed. These measures are necessary, since only once the global prevalence of SARS-CoV-2 infections is reduced substantially can the risk of outbreaks in low prevalence settings be eliminated.

**Reporting summary**. Further information on research design is available in the Nature Research Reporting Summary linked to this article.

## Data availability
All data required to reproduce the results presented here are available at https://github.com/robin-thompson/Outbreaks_Low_Prevalence_Settings/[64]. Data on the proportion of the population vaccinated was obtained from [34] and [35] (see Methods). Source data for the main figures in the manuscript can be accessed at https://github.com/robin-thompson/Outbreaks_Low_Prevalence_Settings/[64]. No restrictions exist on data availability.

## Code availability
All code required to reproduce the results presented here are available at https://github.com/robin-thompson/Outbreaks_Low_Prevalence_Settings/, along with relevant documentation[64].

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

## Acknowledgements

We thank members of the Wolfson Centre for Mathematical Biology (University of Oxford) and the Zeeman Institute for Systems Biology and Infectious Disease Epidemiology Research (University of Warwick) for helpful discussions about this work. We also thank the two reviewers (Pieter Trapman and one anonymous reviewer), as well as Ed Hill and Nick Lai, for helpful feedback that enabled us to improve this manuscript. RS-P was supported by the EPSRC through the Centre for Doctoral Training in Industrially Focussed Mathematical Modelling at the University of Oxford (grant EP/L015803/1), in

collaboration with Biosensors Beyond Borders Ltd. LD was supported by the MRC through the COVID-19 Rapid Response Rolling Call (grant MR/V009761/1) and by UKRI through the JUNIPER modelling consortium (grant MR/V038613/1). This work was funded in part by the UKRI (grant EP/V053507/1). The funders had no role in study design, data collection and analysis, preparation of the paper, or the decision to publish.

## Author contributions

Conceptualisation: RNT, RS-P, LD; Methodology: RNT, RS-P; Investigation: RS-P; Supervision: RNT, HMB; Writing – original draft: RNT, RS-P; Writing – review and editing: All authors.

## Competing interests

The authors declare no competing interests.
