## [Peer Review File · Communications Medicine]

Reviewers' comments:

Reviewer #1 (Remarks to the Author):

COMMSMED-21-0271-T

#Work/Reviewer report/Communications Medicine#

The authors used a branching model to assess the risks of initiating local COVID-19 outbreaks by imported cases after vaccine rollout and the removal of NPIs. Findings from this study have important implications in informing the planning the exit strategy in post-COVID-19 vaccine era, especially for places adopted strict border controls and very low level of COVID-19 circulation during the past year (e.g., Asia and Pacific regions). Overall, this is an impressive work and I only have some minor comments for them to consider (in order of appearance).

1. Lines 27-28: indicate whether these are confidence interval or credible intervals. Also, I think two digits may be good enough for presentation.
2. Lines 51-58: The authors may consider to include the findings that imported cases can contribute considerably to local incidence for those low prevalence regions (Russell TW *et al.*, 2020 *Lancet Global Health*), to emphasise the importance of this study for locations with strict border controls and low COVID-19 prevalence.
3. Lines 63-66: I am surprised that the vaccine effectiveness against infection was as high as 85%. I looked at the reference 17 and the original articles cited in the reference. It seemed that the VE of AZ against asymptomatic infections was 22.2% (-9.9 to 45.0%) (Voysey M. *et al.*, 2020 *Lancet*). Although I believe the lower VE in preventing prevention will not change the conclusion of this study, I hope that a more realistic assumption can be assumed.
4. Lines 80-82: I wonder how did the authors account for super-reading in their model. As the equation indicate, the probability of outbreak is influenced by reproduction number, which is a population average of transmission. However, due to the heterogeneity in COVID-19 transmission, super-reading can happen even when reproduction number was low (Adam DC et al, 2020 *Nature Medicine*).
5. Lines 103-104: The authors may consider to indicate that not the entire population are eligible for vaccination, e.g., young children and the very old elderly for some vaccines. This means that there will be different maximum vaccination uptake proportions for different countries and regions, depending on their population structure and the vaccines they adopted. It would be great if the authors can introduce this constrain. Otherwise, the current wording seems to indicate that all individuals are eligible for vaccination.
6. Lines 174: How was the the maximum proportion v estimated?
7. After reading the methods, I am not very clear how were the removal NPIs were modelled? Did the authors consider the reproduction was only affected by population immunity level, while without any border controls (e.g., testing and quarantines) and social distancing?
8. Lines 204-205: indicate what $q(t)$ denotes.
9. Lines 259-262: It looks like the outbreak risks was calculated by integrating the $R_v(t)$ over time, which means that the choice of study duration may affect the outbreak risks. For example, the if the beginning period of vaccination programme was included, higher $R_v(t)$ were included and may increase the overall outbreak risks. I am wondered, what if the authors set the duration to 3-6 months after the vaccination programs completed, will the outbreak risks change?
10. Lines 322 - may be worthy indicating the exact number for the high vaccine uptake

Reviewer #2 (Remarks to the Author):

This manuscript on the probability of a new SARS-CoV-2 outbreak in the presence of an ongoing vaccination campaign is very well written and the questions answered are relevant at the moment.

However, I do have some important concerns about the paper. The main issue is that equation (1) is very much model dependent. In general there is no direct relationship between R and the probability of a large outbreak. Equation (1) holds for (Markov) SIR epidemics in a homogeneously mixing population with exponentially distributed infectious periods. In general the probability of a large outbreak is given by $1-q$, where q is the smallest solution of $q = \sum p_k q^k$ where p_k is the probability that an infected individual infects k other infected individuals.

In the case of the Markov SIR epidemic the p_k 's describe a geometric distribution with expectation R . However the q is very sensitive to p_k for small k , and I think there are good reasons to assume that there are many infected people who will infect nobody else (even in the absence of behavioral change in response to an outbreak). R (which is equal to $\sum k p_k$) is very much influenced by p_k for large k , (which might be seen as the probability of super spreading events) which is probably larger than what a geometric distribution would predict for SARS CoV2.

So a general rule is R depends mainly on the details of the "many infections tail" of the distribution, while the probability of a large outbreak depends mainly on the low transmission probabilities.

The disruption of the relation between R and the probability of a large outbreak becomes even worse if there are several groups in the population which have different social behavior and therefore different infection rates (and perhaps removal rates).

Now it is definitely true that the results of your paper can be a nice illustration of the power of branching processes in a varying environment, but I do not think that you can get more out of it than crude qualitative insight (which is worth a lot). At the moments the results are presented in a quantitative way that is not justified I think, because of the problems I pointed out in the previous paragraph. This problem does not disappear if the model is complicated by having an R changing over time.

Some detailed comments

L119: The mean does not define the distribution of the infectious period. It should be made more explicit that you assume that the distribution is exponential or that there is a constant rate of recovery (which is suggested in Figure 1, but should come earlier).

L204: I think that in the context used the paper by David Kendall

"On the generalized" birth-and-death" process." The annals of mathematical statistics 19.1 (1948): 1-15.

is very relevant and should be referred to (although it probably happens indirectly through the already present references). However, also this paper only deals with a situation where the dynamics can be described by infection and removal rates (albeit here time varying) which do not depend on the time since infection.

L375: Are the references 50 and 51 still the best information available? I read some news flashes that there is now more evidence that immunity (through vaccination or previous infection) may last long (though not for everybody).

To summarize, I like the paper very much, but the first problem outlined above is makes that I

recommend against publication of the manuscript in the present form. If there is much stress from the start that the results should be interpreted qualitatively and the issues I pointed out are discussed (perhaps in an appendix) then I think the manuscript is worth publishing (not many people in the applied field know how to deal with varying "environments"), but I leave it to the editors whether the results are interesting enough for this journal.

Please note: In the material below, whenever we refer to line numbers in the revised manuscript, these refer to the version of the manuscript with “track changes”. We have uploaded two versions of the revised manuscript – one with, and one without, track changes.

Reviewer 1

The authors used a branching model to assess the risks of initiating local COVID-19 outbreaks by imported cases after vaccine rollout and the removal of NPIs. Findings from this study have important implications in informing the planning the exit strategy in post-COVID-19 vaccine era, especially for places adopted strict border controls and very low level of COVID-19 circulation during the past year (e.g., Asia and Pacific regions). Overall, this is an impressive work and I only have some minor comments for them to consider (in order of appearance).

Response: *Thank you for recognising the important implications of our research, and for the minor comments that have helped us to improve our manuscript.*

1. Lines 27-28: indicate whether these are confidence interval or credible intervals. Also, I think two digits may be good enough for presentation.

Response: *These are 95% equal-tailed credible intervals (since we are taking a Bayesian approach). We have now amended the main text so that all results are stated to two significant figures, and clarified that we are using 95% equal-tailed credible intervals (lines 354-357).*

2. Lines 51-58: The authors may consider to include the findings that imported cases can contribute considerably to local incidence for those low prevalence regions (Russell TW *et al.*, 2020 *Lancet Global Health*), to emphasise the importance of this study for locations with strict border controls and low COVID-19 prevalence.

Response: *Thank you for this excellent suggestion. We have now noted the role of imported cases in driving local incidence in the revised manuscript (lines 65-68), and cited the article by Russell *et al.* We agree with the reviewer that this emphasises the importance of our study.*

3. Lines 63-66: I am surprised that the vaccine effectiveness against infection was as high as 85%. I looked at the reference 17 and the original articles cited in the reference. It seemed that the VE of AZ against asymptomatic infections was 22.2% (-9.9 to 45.0%) (Voysey M. *et al.*, 2020 *Lancet*). Although I believe the lower VE in preventing prevention will not change the conclusion of this study, I hope that a more realistic assumption can be assumed.

Response: *This is a really interesting point. The reviewer is correct that assuming a lower VE does not change the conclusion of our study (in fact, it*

leads to higher outbreak risks which further emphasise the need to maintain surveillance for infected inbound travellers once NPIs are removed).

There is substantial uncertainty in the extent to which two vaccine doses prevent infection by SARS-CoV-2, and this is likely to be affected by a range of factors (including the precise vaccine considered and the time interval after vaccination at which vaccine efficacy is quantified). However, it is now widely accepted that two vaccine doses provide high protection against SARS-CoV-2 infection. The value of 85% (two weeks after the second vaccine dose) was taken from the UK government's COVID-19 roadmap modelling. A recent CDC study in fact found a higher value of 90% protection after two vaccine doses (<https://www.bmj.com/content/373/bmj.n888>).

Interestingly, recent reports from Israel seem to suggest a lower vaccine effectiveness at preventing infections by the Delta variant (64% - down from estimates of 94% for previous variants). It remains to be seen whether these reports are substantiated by additional data. However, in any case, the overall protection against infection is likely to be substantially higher than 22.2%.

Nonetheless, to reflect the uncertainty in the protection afforded by two vaccine doses, we have conducted a supplementary analysis in which we show that our main conclusion is robust to the assumed value of η_2 (Fig S3). As described above, lower assumed VE values further emphasise our conclusion: the outbreak risk is not eliminated completely in low prevalence settings following vaccination programmes.

4. Lines 80-82: I wonder how did the authors account for super-reading in their model. As the equation indicate, the probability of outbreak is influenced by reproduction number, which is a population average of transmission. However, due to the heterogeneity in COVID-19 transmission, super-reading can happen even when reproduction number was low (Adam DC et al, 2020 *Nature Medicine*).

Response: *The reviewer is correct that heterogeneity in transmission between different infected individuals influences the risk of outbreaks. The potential for superspreading is not included in simple branching process models (which assume a geometric offspring distribution). However, to explore how the potential for superspreading affects our main conclusion, we have included a new supplementary analysis in our revised manuscript (see Fig S4 and the Supplementary Information section titled "Individual-level variation in SARS-CoV-2 transmission"). This demonstrates that our main conclusion holds when the potential for superspreading is included in the underlying branching process model (please also see response to Reviewer 2 for further details).*

5. Lines 103-104: The authors may consider to indicate that not the entire population are eligible for vaccination, e.g., young children and the very old elderly for some vaccines. This means that there will be different maximum vaccination uptake

proportions for different countries and regions, depending on their population structure and the vaccines they adopted. It would be great if the authors can introduce this constrain. Otherwise, the current wording seems to indicate that all individuals are eligible for vaccination.

Response: *Thank you for the opportunity to clarify the wording here. In the model, we included the feature that the entire population is not eligible for vaccination in our analysis via the parameter v . This reflects the population structure of the locations concerned (the higher value of v for the Isle of Man compared to Israel reflects the fact that there are more adults there, who are more likely to be vaccinated than children). We hope that this is now clearer with the revised wording (see, for example, lines 125-127).*

6. Lines 174: How was the maximum proportion v estimated?

Response: *We took estimates of v from the literature (reference 43 in our manuscript), and have cited the relevant article in Table 1. It is also interesting to examine recent vaccination data from those countries, which support our assumed values (around 65% of all individuals in Israel have now been vaccinated, and around 75% in the Isle of Man, which are in line with final assumed vaccination uptake values of 70% for Israel and 80% for the Isle of Man). The different values of v in different locations reflect the structures of the host populations, as described above.*

7. After reading the methods, I am not very clear how were the removal NPIs were modelled? Did the authors consider the reproduction was only affected by population immunity level, while without any border controls (e.g., testing and quarantines) and social distancing?

Response: *The goal of our manuscript was to assess the potential for outbreaks following the removal of all travel restrictions and other NPIs. As a result, we considered baseline mean R_0 values of 3 (e.g. Figs 3a, 3c) and 5 (e.g. Figs 3b, 3d) in the absence of NPIs, and then adjusted those values to include vaccine-acquired immunity due to the vaccination programmes. The value of $R_0=3$ is consistent with early estimates for SARS-CoV-2, and the value of $R_0=5$ reflects the increased transmissibility of novel variants. By considering the potential for outbreaks in the absence of border controls, we emphasise the need to continue to implement some measures (e.g. surveillance of imported cases) even when most interventions have been relaxed. We have now included additional text to clarify that we are considering scenarios in which transmission is limited by vaccine-acquired immunity alone via an adjustment of R_0 to $R_v(t)$ (lines 150-151, 229-231).*

8. Lines 204-205: indicate what $q(t)$ denotes.

Response: *We have now addressed this in the main text, as requested (lines 252-255).*

9. Lines 259-262: It looks like the outbreak risks was calculated by integrating the $R_v(t)$ over time, which means that the choice of study duration may affect the outbreak risks. For example, if the beginning period of vaccination programme was included, higher $R_v(t)$ were included and may increase the overall outbreak risks. I am wondered, what if the authors set the duration to 3-6 months after the vaccination programs completed, will the outbreak risks change?

Response: *Thank you for the opportunity to clarify this. Rather than integrating $R_v(t)$ over time, we instead calculate the outbreak risk at time t accounting for *all* future changes in the numbers of vaccinated hosts. Thus, the risk of an outbreak based on an imported case arriving at time t depends on the number of vaccinated hosts at all times after time t . As a result, it is not required to set a study duration (all times after the end of the vaccination programme are already included in our calculation).*

The only integration that we perform is to integrate the outbreak risk at time t over the current distributional estimate of $R_v(t)$ – in other words, the integration is over possible values of R_v , rather than over values of t . The purpose of this is to account for uncertainty in the precise value of $R_v(t)$. We have clarified this in the text (lines 324-330).

10. Lines 322 - may be worthy indicating the exact number for the high vaccine uptake

Response: *We have added this to the manuscript, as suggested.*

Thank you again for your very helpful suggestions, which have helped us to improve our manuscript.

Reviewer 2

This manuscript on the probability of a new SARS-CoV-2 outbreak in the presence of an ongoing vaccination campaign is very well written and the questions answered are relevant at the moment.

Response: *Thank you very much for these positive comments, and for the very useful suggestions below. We have addressed these comments here and in the revised manuscript.*

However, I do have some important concerns about the paper. The main issue is that equation (1) is very much model dependent. In general there is no direct relationship between R and the probability of a large outbreak. Equation (1) holds for (Markov) SIR epidemics in a homogeneously mixing population with exponentially distributed infectious periods. In general the probability of a large outbreak is given by $1-q$, where q

is the smallest solution of $q = \sum p_k q^k$ where p_k is the probability that an infected individual infects k other infected individuals.

In the case of the Markov SIR epidemic the p_k 's describe a geometric distribution with expectation R . However the q is very sensitive to p_k for small k , and I think there are good reasons to assume that there are many infected people who will infect nobody else (even in the absence of behavioral change in response to an outbreak). R (which is equal to $\sum k p_k$) is very much influenced by p_k for large k , (which might be seen as the probability of super spreading events) which is probably larger than what a geometric distribution would predict for SARS CoV2.

So a general rule is R depends mainly on the details of the "many infections tail" of the distribution, while the probability of a large outbreak depends mainly on the low transmission probabilities.

The disruption of the relation between R and the probability of a large outbreak becomes even worse if there are several groups in the population which have different social behavior and therefore different infection rates (and perhaps removal rates).

Now it is definitely true that the results of your paper can be a nice illustration of the power of branching processes in a varying environment, but I do not think that you can get more out of it than crude qualitative insight (which is worth a lot). At the moments the results are presented in a quantitative way that is not justified I think, because of the problems I pointed out in the previous paragraph. This problem does not disappear if the model is complicated by having an R changing over time.

Response: *The reviewer is correct that the precise quantitative outbreak risk estimates depend on assumptions made in the underlying transmission model, including the potential for superspreading (and heterogeneities in transmission between different infectious groups).*

However, our main conclusion – that the outbreak risk is not eliminated in low prevalence settings even when vaccination programmes are completed, thereby emphasising the need for continued vigilance when NPIs are lifted – is robust to whether or not the “many infections tail” is included in the underlying transmission model.

To demonstrate this, we have conducted a new supplementary analysis in which we consider the outbreak risk at the end of the vaccination programmes in the Isle of Man and Israel (see Supplementary Information section titled “Individual-level variation in SARS-CoV-2 transmission” and Fig S4; and lines 535-545 of the main text). We extend the standard branching process model used in the main text to include p_k 's (the offspring distribution) that are governed by a negative binomial distribution. We consider a range of values of the dispersion parameter of the negative binomial distribution (where small values – $k < 0.5$ – have previously been estimated for SARS-CoV-2).

In this new supplementary analysis, we demonstrate clearly that our main conclusion holds when the potential for superspreading is included in the branching process model. Specifically, the outbreak risk is greater than zero once vaccination programmes are completed, even when a negative binomial offspring distribution is used. This emphasises the need for ongoing surveillance when travel restrictions are removed.

Some detailed comments

L119: The mean does not define the distribution of the infectious period. It should be made more explicit that you assume that the distribution is exponential or that there is a constant rate of recovery (which is suggested in Figure 1, but should come earlier).

Response: *We have added clarifying text at the start of the Methods, as suggested. We also now noted in the manuscript that Equation (1) is based on assumptions of homogeneous mixing and exponentially distributed infectious periods.*

L204: I think that in the context used the paper by David Kendall "On the generalized" birth-and-death" process." The annals of mathematical statistics 19.1 (1948): 1-15. is very relevant and should be referred to (although it probably happens indirectly through the already present references). However, also this paper only deals with a situation where the dynamics can be described by infection and removal rates (albeit here time varying) which do not depend on the time since infection.

Response: *Thank you for referring us to the excellent paper by Kendall. As the reviewer suggested, we now refer to this paper in the revised manuscript (line 254).*

L375: Are the references 50 and 51 still the best information available? I read some news flashes that there is now more evidence that immunity (through vaccination or previous infection) may last long (though not for everybody).

Response: *This is a really interesting point. The extent to which immunity wanes through time is still very uncertain. While antibodies may decrease over time, it is unclear precisely what implications this may have on the immune response. It is hoped that cellular immune responses may be maintained for long periods, leading to long-lasting immune memory of previous SARS-CoV-2 infections, at least for some individuals.*

There is also uncertainty about the duration over which vaccines provide protection, and booster vaccinations are currently being considered. However, we note that we chose to make the assumption that vaccine-acquired immunity does not wane. Any waning of immunity will act to strengthen our main conclusion (that the outbreak risk is non-zero following removal of NPIs). We

have added discussion of the above issues to the revised manuscript (lines 505-514).

To summarize, I like the paper very much, but the first problem outlined above is makes that I recommend against publication of the manuscript in the present form. If there is much stress from the start that the results should be interpreted qualitatively and the issues I pointed out are discussed (perhaps in an appendix) then I think the manuscript is worth publishing (not many people in the applied field know how to deal with varying "environments"), but I leave it to the editors whether the results are interesting enough for this journal.

Response: *Thank you for your positive comments and constructive feedback on our manuscript. As described above, we have now conducted a supplementary analysis in which we explore the effects of heterogeneity in transmission between different infectious hosts, and we show that our main conclusion is unchanged. In addition, we have now emphasised throughout our manuscript that our main results intend to demonstrate the key general principle that outbreak risks are not eliminated completely following vaccination programmes in low prevalence settings, rather than to generate exact quantitative predictions of this outbreak risk (e.g. lines 23-27, 122-129, 357-360, 426). We have also removed the quantitative results from the Abstract to reflect this. We think that these changes have improved our manuscript significantly.*

Thank you again for your very helpful suggestions, which have helped us to improve our manuscript.

REVIEWERS' COMMENTS:

Reviewer #1 (Remarks to the Author):

The authors have addressed all my comments. I have no other comments.

Reviewer #2 (Remarks to the Author):

This second version of the manuscript is a considerable improvement compared to the first version. Mathematically I do not have much to add to it. Whether the results are interesting enough for publication, I feel less qualified to comment on than the editors are.

There are some minor comments that I still might be addressed.

The first is on the last line of the abstract "...should remain in place". I think this "should" assumes that as low spread as possible within the country (or Island) is the ultimate target. I doubt whether this is the case and whether global case numbers can ever be reduced to ever lift this measure. I personally think that there will be a moment that governments will say: the inconvenience and cost of the measures is more than what we value the gain of keeping the disease out of our mostly protected country. Anyway, my remark is that you should soften "should" or be more specific about what the goal is for which some measures should stay in place in order to achieve it.

After equation (1) you specify what model assumptions are needed for (1) to hold true, but still it reads as if (1) holds more general. I suggest to add in line 92 something along the lines of "...branching processes, for the most common epidemic model the probability" and further leave the paragraph mostly as it is. Or actually add that the theory on branching processes gives for very general models that the probability of a new introduction leading to a large outbreak is strictly positive if and only $R > 1$ (or actually when the ultimate R exceeds 1). This holds for all reasonable epidemic models for SARS-cov2 (Mathematicians have to put effort in creating a branching process for which this "equivalence" does not hold), and it might be worth mentioning that. Your further work in this manuscript still gives a good qualitative insight in how introduction probabilities behave in a reasonable model.

Line 475/476: I do not like "best case" here, even in parantheses it is too strong I think (See your reference 61 by Britton, Ball and Trapman.

Pieter Trapman

Reviewer 1

The authors have addressed all my comments. I have no other comments.

Response: *Thanks again for your very helpful comments, which helped us to improve our manuscript.*

Reviewer 2

This second version of the manuscript is a considerable improvement compared to the first version. Mathematically I do not have much to add to it. Whether the results are interesting enough for publication, I feel less qualified to comment on than the editors are.

There are some minor comments that I still might be addressed.

Response: *Thanks for helping us to improve our manuscript, and for the additional minor comments that we address below.*

The first is on the last line of the abstract "...should remain in place". I think this "should" assumes that as low spread as possible within the country (or Island) is the ultimate target. I doubt whether this is the case and whether global case numbers can ever be reduced to ever lift this measure. I personally think that there will be a moment that governments will say: the inconvenience and cost of the measures is more than what we value the gain of keeping the disease out of our mostly protected country. Anyway, my remark is that you should soften "should" or be more specific about what the goal is for which some measures should stay in place in order to achieve it.

Response: *We agree with the reviewer, and have rephrased this statement. Rather than asserting that surveillance measures should remain in place, we instead note that such measures are useful tools for suppressing potential outbreaks.*

After equation (1) you specify what model assumptions are needed for (1) to hold true, but still it reads as if (1) holds more general. I suggest to add in line 92 something along the lines of "...branching processes, for the most common epidemic model the probability" and further leave the paragraph mostly as it is. Or actually add that the theory on branching processes gives for very general models that the probability of a new introduction leading to a large outbreak is strictly positive if and only $R > 1$ (or actually when the ultimate R exceeds 1). This holds for all reasonable epidemic models for SARS-cov2 (Mathematicians have to put effort in creating a branching process for which this "equivalence" does not hold), and it might be worth mentioning that. Your further work in this manuscript still gives a good qualitative insight in how introduction probabilities behave in a reasonable model.

Response: *As the reviewer has suggested, we now note that the probability that introductions lead to an outbreak driven by sustained local transmission is greater than zero whenever R is greater than one. We then refer to equation (1) as the outbreak probability for the most common branching process model.*

Line 475/476: I do not like "best case" here, even in parantheses it is too strong I think (See your reference 61 by Britton, Ball and Trapman.

Response: *We have removed "best case" from the manuscript.*